# Can Neural Networks Learn Implicit Logic from Physical Reasoning?

**Aaron Traylor,**[1] **Roman Feiman,**[2] **& Ellie Pavlick**[1]
[1] Department of Computer Science
[2] Department of Cognitive, Linguistic & Psychological Sciences
Brown University, Providence, Rhode Island, USA
`{aaron_traylor, roman_feiman, ellie_pavlick}@brown.edu`

## Abstract

Despite the success of neural network models in a range of domains, it remains an open question whether they can learn to represent abstract logical operators such as *negation* and *disjunction*. We test the hypothesis that neural networks without inherent inductive biases for logical reasoning can acquire an *implicit* representation of negation and disjunction. Here, *implicit* refers to limited, domain-specific forms of these operators, which work in psychology suggests may be a precursor (developmentally and evolutionarily) to the type of abstract, domain-general logic that is characteristic of adult humans. To test neural networks, we adapt a test designed to diagnose the presence of negation and disjunction in animals and pre-verbal children, which requires inferring the location of a hidden object using constraints of the physical environment as well as implicit logic: if a ball is hidden in A or B, and shown not to be in A, can the subject infer that it is in B? Our results show that, despite the neural networks learning to track objects behind occlusion, they are unable to generalize to a task that requires implicit logic. We further show that models are unable to generalize to the test task even when they are trained directly on a logically identical (though visually dissimilar) task. However, experiments using transfer learning reveal that the models do recognize structural similarity between tasks which invoke the same logical reasoning pattern, suggesting that some desirable abstractions are learned, even if they are not yet sufficient to pass established tests of logical reasoning.

## 1 Introduction

People have the capacity for flexible logical reasoning. For example, given two alternatives (A or B), and subsequent information that allows ruling out one of them (not A), people can conclude that the other is true with certainty (therefore B), termed *reasoning by exclusion*. It is an open question whether achieving similar reasoning with neural models will require explicit logical components to be built into the network architecture or if the capacity for such reasoning can be learned from data.

Prior work on logical reasoning in neural networks (Marcus, 2001; Evans et al., 2018, among others) has focused on whether models are able to acquire abstract, domain-general logical operators, such as the ¬ and ∨ found in first order logic. However, recent psychological studies of logic in non-human animals and human infants have suggested that this powerful reasoning machinery does not appear in adults fully formed, *ex nihilo*. Rather, this work has argued that both over the human lifespan and across species, domain-general logical operators may develop and evolve from scaffolding provided by precursors that are themselves more limited. These precursors are *implicit* logical operators that can differ from the explicit, domain-general forms in two ways: they might only be able to operate on content in specific domains, and they might perform only some of the functional role of their full-fledged counterparts (Völter & Call, 2017; Bermudez, 2003; Cesana-Arlotti et al., 2018), (see Feiman et al., 2022, for discussion).

In this work, we ask whether neural network models, which lack explicit representations of the logical operators for negation and disjunction, can nonetheless acquire implicit representations of such operators via self-supervised training. In particular, we focus on implicit logic within the domain

of intuitive physics, as this is one of the earliest domains in which such reasoning emerges in young children (Cesana-Arlotti et al., 2018; 2020; Feiman et al., 2022). We design a set of experiments in which models are trained to track objects as they move throughout a visual scene, and then evaluated on a task from developmental psychology that requires reasoning about the location of a hidden object and is considered to be a face-valid test for the representation of (implicit) negation and disjunction. We find that, by most measures, object-tracking neural networks are unable to generalize zero-shot to the logical reasoning test, even when given training data which directly illustrates the requisite reasoning pattern. However, in transfer learning experiments, we find evidence that neural networks encode some degree of structural similarity between visually distinct but logically equivalent tasks, suggesting that they may yet be capable of representing the desired operators. Future work will need to determine the exact training conditions under which they would do so.

In summary, our primary contributions are: (1) We introduce the notion of *implicit* logic, taken from developmental and comparative psychology, into the repertoire of neural network evaluation; (2) We adapt a standard test of logical inference in humans and use it to evaluate neural network models; (3) We present a series of studies which present primarily negative results regarding neural networks' ability to learn implicit logical reasoning in the physical domain, but which offer some suggestive evidence regarding the models' ability to transfer representations between logically equivalent tasks.

## 2 BACKGROUND

### 2.1 TWO TESTS OF REASONING BY EXCLUSION

To test whether computational models can reason using implicit negation and disjunction, we adapt two tasks previously used with infants (Feiman et al., 2022; Piaget, 1954) and many species of non-human animals (Call, 2004; Völter & Call, 2017, for review). In the "Two-Cup" task (see Figure 1), participants first see two cups, which are then hidden behind a screen. An object (e.g., a toy or food) is then lowered behind the screen into one cup (setting up A or B). The screen is then removed, showing that one cup is empty (not A), licensing the inference that the object must be in the other cup (therefore B). Finally, participants are invited to search. Success requires representing (explicitly or implicitly) that the ball is *not* in the empty cup in order to avoid searching there. Infants and many animal species succeed on this task in a zero-shot setting. In Piaget's (1954) "Invisible Displacement" paradigm, participants see a hand holding an object. The hand closes to hide the object, moves behind an occluder, and then emerges again, empty palm facing the participant (A or B; not A). This licenses the inference that the object must have been deposited behind the occluder (therefore B). In this work, we use the Two-Cup task as our target test task. In Sections §4.2 and §4.3, we train on Invisible Displacement in order to assess whether a neural model trained to solve one reasoning-by-exclusion task can transfer its representations to another task that is formally similar but visually distinct.

### 2.2 EXPLICIT VS. IMPLICIT LOGICAL REASONING

One way to solve both tasks is with *explicit* symbolic logical reasoning: represent the initial possibilities for the object's location as A∨B, and represent evidence ruling out one of them as ¬A, thus licensing the conclusion B. However, more minimal solutions are also possible. Feiman et al. (2022) propose that logical representations (negation, disjunction), can be *implicit* in two senses. First, while explicit logic is characteristically domain-general (OR and NOT can compose with any concepts, regardless of their content), implicit logical operations can be domain-specific, operating only over certain kinds of content (e.g. representations of objects' locations). Second, implicit logical representations might perform only part of the function of their explicit counterparts. For example, a function that compares two arguments (e.g. *blue* and *red*) for incompatibility is an implicit negation in this sense. It plays part of the functional role of representing *blue* as *not red* even as it would (correctly) not represent the negation of *red* as equivalent to *blue* in other computational contexts. In the Two-Cup task, participants could use an implicit representation of negation to represent that the cup being *empty* is incompatible with it *containing the object* (see Feiman et al., 2022, for discussion).

After ruling out the empty cup from consideration, further deriving the certain conclusion that the object *must* then be in the other cup is a signature of *disjunction*. This conclusion could be licensed by an explicit logical operator (A∨B), but it could also be the consequence of an implicit representation with only part of the functional role of ∨. For instance, the two options could be linked

probabilistically, such that learning that the object is *not* in one cup increases the probability that it is in the other (see Feiman et al., 2022; Mody & Carey, 2016; Rescorla, 2009).

## 2.3 THIS WORK

(Feiman et al., 2022) propose that such implicit logical representations – both with partial function and limited to a specific domain – could be evolutionary and developmental precursors to their explicit counterparts. On this account, implicit logical representations could emerge from specific content knowledge that is not logically structured, such as the intuitive physical understanding that predicts how objects might move in space based on visual input. In this paper, we test whether models trained to represent object-tracking and occlusion will possess emergent representations of negation and disjunction that are implicit in both senses: representations limited to the content domain of intuitive physics, and to the function of detecting incompatibility between two input states.

## 3 EXPERIMENTAL DESIGN

### 3.1 EVALUATION TASK

To test whether a model has an implicit representation of negation and disjunction, we evaluate its performance on the Two-Cup task. Figure 1 shows our implementation of this task. We convert the task into a 2D video format using `pygame` and `Box2D`. Our environment consists of wedges and cups (which are static), a ball (which obeys gravity), and occluders (which move up and down from the bottom of the screen). The task is to predict the location of the ball at each frame of the video. In each sequence, the ball falls from the top of the screen, and can either roll off the wedge into the left or the right cup, each of which are occluded. Because the ball's path off the wedge is also occluded, it is ambiguous which cup the ball falls into until the occluders are removed. The occluder in front of the cup that does not contain the ball is the first to fall, thus licensing the negation inference.

We designate the frame at which the first cup is revealed to be empty (frame `f` in Figure 1) to be the *critical frame*–i.e., the frame at which a model with implicit logic should be able to determine with certainty the location of the ball. We thus evaluate models by comparing before and after the critical frame whether the model predicts the ball to be in the correct cup, the incorrect cup, or elsewhere on the screen. After our initial tests showed that variance in performance across examples was low, we constructed a set of 100 test examples over which to compute this metric.

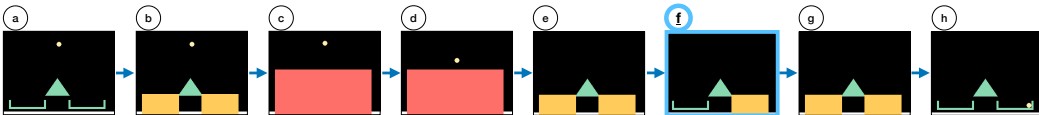

Figure 1: Frames from the Two-Cup task test video. Frames `a-d` create ambiguity between the two possible hiding locations for the ball, and frames `e-h` are the part of the task on which we evaluate the model. The goal is to determine the location of the small yellow ball, which drops at frame `d`. At frame `e`, the subject should know that the ball is either in the left cup or the right cup, but its exact location is unknown. When the left occluder drops at frame `f`, the subject should know that the ball is *not* in the left cup, and *must* be in the right cup, despite not seeing it. This is the critical frame.

### 3.1.1 MODEL

To test whether the capacity for implicit logical reasoning emerges in systems designed to represent an intuitive understanding of Newtonian physics, we use a neural network model that has a built-in notion of discrete objects, but which must learn the physical dynamics (and their generalization to logical reasoning tasks) from data. In addition to being consistent with developmental evidence on the emergence of logic, this architectural choice simplifies the learning problem and thus allows us to study the phenomena we aim to study without needing to train extremely large models, as would be the case if we took a tabula rasa approach.

We use the Object-Permanence Network (OPNet) model described by (Shamsian et al., 2020). OPNet is an LSTM trained on video data that tracks objects as they move and are occluded by other objects. It thus learns to track and predict object locations without a built-in representation of object physics. Specifically, OPNet consists of **a)** an object detection module that segments a frame into visible elements (rather than use this module, we feed in ground truth bounding boxes), **b)** an LSTM that attends to the scene and produces a distribution over current objects to identify which the target object (in our case, the ball) is currently behind, if any, and finally **c)** a "where is it" LSTM that predicts a bounding box for the target object based on a weighted average of components a) and b). [1]

### 3.1.2 TRAINING

In our experiments (§4.1,4.2,4.3), we consider a variety of training conditions in order to determine which, if any, allow the model to encode an implicit negation and disjunction to succeed on the Two-Cup task. In all experiments, we require at a minimum that the model has been exposed to the basic elements of the Two-Cup task, so that the visual appearance of this task is in-distribution. We also require that the evaluation task itself (namely, the sub-sequence of frames (e)–(h) in Figure 1) remains entirely unseen in training, to ensure that the model cannot exploit specific heuristics of that sequence in order to solve the task. With these constraints, we design three sets of training data schemas, described below in Section 4. Each training set contains 5,000 videos of 57 frames each.

**Physical Reasoning Schema** We generate simple scenes which show the basic physical phenomena of the environment. Each example has a randomly selected template, each of which enumerates different interactions the ball may have within the environment; e.g. it falls into the cup, it falls onto a wedge and then into a cup, it falls onto a wedge and then rolls along the ground, etc. In each template, the ball falls from a randomly chosen point on the x-axis (though the height from which it falls is kept constant) at a random time within the first 40 frames of the video. Randomly generated occluders appear throughout the scene and may or may not block the path of the ball. Attributes of the cups and wedges (e.g. x-location, size, height) are also chosen uniformly at random from pre-set ranges. When the ball rolls along the ground of the scene, it will stop at a randomly selected x location [2]

**Two-Cup Ablations Training Schema** We construct a set of training examples which are designed to expose the model to the general concepts necessary to solve the Two-Cup task, without giving it access to heuristics it could use to succeed on evaluation without representing implicit logic. Specifically, we considered all variations and perturbations of the Two-Cup task's sequence of events, e.g., varying the number of cups, number of occluders, and the order in which the occluders rise and fall. We then filter out any sequence of events which either (1) were logically equivalent to the Two-Cup task and thus would prevent the Two-Cup tasks from being held out or (2) had direct overlap with the true Two-Cup task's frames at and after the critical frame, even if they did not require the same logical reasoning. For example, if in Figure 1 we changed the order of the events in frames `b-d` from "small occluder rise", "large occluder rise", "ball fall" to "ball fall", "small occluder rise", "large occluder rise" (`d-b-c`), it would not be permissible in this training set because frames `e-h` remain the same as the Two-Cup task. We found that 38 perturbations of the Two-Cup Scene were permissible, and a full list is available in the Appendix. For each video made using this procedure, we randomly select a permissible perturbation, and vary elements such as cup size, wedge size, and x-axis of the midpoint of the scene, but keep the order of events and location of the ball constant.

**Invisible Displacement Training Schema** Finally, we generate a training set which exposes the model to the same logical reasoning pattern that occurs in the Two-Cup task, but which is visually distinct. We use this training data in a subset of our experiments (§4.2) in which we seek to provide the model with a better chance at passing the Two-Cup task without training on the Two-Cup task itself. To generate such a training set, we use a variation on Piaget's Invisible Displacement experiments (§2.1). In our Invisible Displacement data, as shown in Fig. 2, the ball rolls off of a wedge and then behind two occluders that are positioned directly next to each other. The ball stops at a random point

---

[1]The loss in the original OPNet model consists of MAE compared to the ground truth location of the ball, and a "consistency error" that penalizes the model for moving its prediction from one frame to the next. In order to ensure that the model can update its predictions during the Two-Cup task, we remove the consistency loss.

[2]This constraint ensures compatibility with the Invisible Displacement training schema. In the examples derived from the Two-Cup task, the ball does not roll along the ground.

behind the occluders (similar to the procedure in the Physical Reasoning schema), and its location is ambiguous until one occluder falls, revealing that the location behind it is empty.

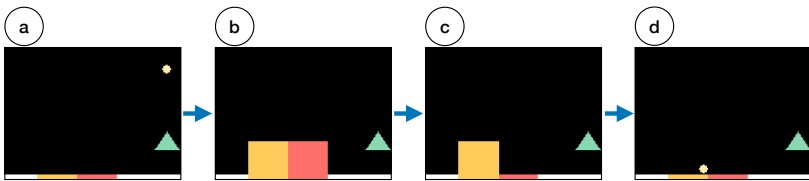

Figure 2: Frames from the Invisible Displacement task within our experimental framework. At frame b, because of the random stopping, it is ambiguous whether the ball has landed behind the yellow or the red occluder. At the critical frame c, it becomes possible to infer the ball's location.

## 3.2 PROOF-OF-CONCEPT

Our primary studies (§4) ask whether neural network models can learn implicit negation and disjunction in the context of physical reasoning tasks. In this section, as a proof-of-concept, we demonstrate that such implicit reasoning is possible. Using a model that has a built-in (rather than learned) physics engine, we illustrate that a model with an explicit representation of physical dynamics *is* able to reason using implicit negation and disjunction, and can succeed on the Two-Cup task.

### 3.2.1 MODEL

We use a pared-down version of ADEPT (Smith et al., 2019), a model of humans' object tracking as constrained by their intuitive physics. This model is not trained and works out-of-the-box. ADEPT uses a Bayesian filtering algorithm to represent beliefs about the locations of occluded objects, and probabilistically represents potential future trajectories of those objects using a built-in physics simulator. This model is not trained and works out-of-the-box. For a video sequence $x_1, x_2, ..., x_t, ...x_T$, the model receives ground truth object segmentations of bounding boxes around the visible objects in the scene at each frame $x_i$. Objects that are occluded are not included.

An important feature of ADEPT is its ability to maintain beliefs about multiple possible worlds, which is done explicitly using a particle filter. ADEPT maintains a set of "particles", each of which is a representation of the complete scene at time $t$. The next step of each particle is generated using a noisy internal physics simulator. After several steps, the particles are resampled based on their likelihood, which is calculated based on the overlap between the particle's representation of the world and the observations. Additional details explaining the ADEPT model are included in the Appendix.

We evaluate the accuracy of the ADEPT model on the Two-Cup task data using 100 particles [3]. We calculate the likelihood of each particle at the critical frame, and then group all particles based by where they represent the ball in the scene. If a particle represents the ball as being in the same cup that the ground truth ball is in, that particle is coded as "correct cup", if it's in the other cup it is coded as "incorrect cup", and anywhere else on the screen is "elsewhere". We then sum the likelihood of the particles in each of the three groups to arrive at a final score.

### 3.2.2 RESULTS AND DISCUSSION

Our results are shown in Figure 3. Before the critical frame, the ADEPT model is uncertain about the location of the ball (frame f), and its belief is distributed roughly evenly over the two cups. After the critical frame, its belief shifts entirely to the correct cup (frame g). This pattern is consistent with the negation and disjunction inferences, and the ADEPT model thus succeeds at the Two-Cup task.

---

[3]We experimented with many values of particle count, and found that the choice did not affect the results as long as there was a reasonable chance that at least one particle would end up in each cup.

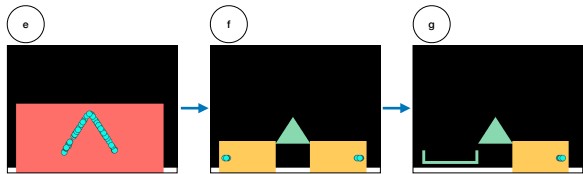

Figure 3: Key frames of the Two-Cup task, visualizing the particles of the ADEPT model. There are 100 particles, and each circle is one particle's representation of its beliefs about the ball's location.

These results illustrate how implicit representations of negation and disjunction can work. The model succeeds at the Two-Cup task, but it does so using a representational mechanism specific to the physical reasoning domain (here, the particle filter). The disjunction is implicit by virtue of all the particles being simulated in either cup at the same time, prior to the reveal of the empty cup. The negation is then implicit in how ADEPT responds to the empty cup. The probability of resampling any particle in which the ball had been in the empty cup is reduced to zero. Clearly, this mechanism could not readily transfer to other logical reasoning tasks (e.g., linguistic reasoning) in the way a formal logical operator could. These results thus illustrate a plausible way in which neural network models might learn to solve the same task by modeling the physical world. Namely, if the neural network learns to represent the physics of objects' motion, as well as a representation of uncertainty and resampling in the face of unexpected observations, it should be able to pass the Two-Cup task.

## 4 EXPERIMENTS

We present three experiments, which get progressively easier, each requiring less generalization from the model than the previous. This experimental structure is due to the fact that we observe negative results in the earlier experiments, and thus progressively relax requirements in order to understand the conditions under which the model can succeed (if at all). First, we train OPNet only on a basic object tracking task, and evaluate whether such training is sufficient for the model to encode an implicit negation and disjunction, as measured by the Two-Cup task (§4.1). Second, we train on the Invisible Displacement task (§2.1) in addition to basic object tracking (§4.2). Finally, we consider a transfer learning paradigm, in which the model is trained first on the Invisible Displacement task and then fine-tuned on the Two-Cup task (§4.3).

### 4.1 EXPERIMENT 1: ZERO-SHOT TRANSFER FROM OBJECT TRACKING

We first test whether a model trained only on object tracking will acquire a capacity for implicit logical reasoning that is sufficient to generalize directly to the Two-Cup task. We train an OPNet model on Physical Reasoning + Two-Cup Ablations data. To compare this model's performance to a plausible ceiling benchmark, we also train a model on a separate dataset of 5,000 Two-Cup task videos, which are instead drawn from the same distribution as the held-out Two-Cup test set. We compare the accuracy of the two models to "hypothetical success" on the Two-Cup task – before the critical frame, the ball could be in either cup, but after the critical frame in which one cup is revealed to be empty, an ideal model will always predict that the ball is in the other cup. We select the model that minimizes loss on the dev set (100 examples for Physical Reasoning + Two-Cup Ablations data, 50 examples for Two-Cup data). Model training details are in the Appendix.

**Control Experiments** In order to ensure that the experimental model has learned to track objects in this environment, we also test it on several control datasets. These control datasets are permutations of the Two-Cup task and are intended to ensure that the model has succeeded in encoding the basic environmental knowledge (other than the logical reasoning patterns) on which the Two-Cup task depends. First, **Two-Cup Freebie** is designed to rule out failure based on the task setup. This test set contains the same setup as Two-Cup; but the first revealed cup shows the ball, thus removing the need to reason by exclusion. Next, to ensure that the model can track the ball's location, even when occluded, we create the **Known Specific Location** and **Known Cup Location** test sets. In the former, we reorder the frames in Fig. 1 such that the ball falls, then the occluders rise, before proceeding like

| Train on Two-Cup task, test on Two-Cup task | | | |
|---|---|---|---|
| Dataset | Correct | Incorrect | Elsewhere |
| Two-cup task | 0.81 | 0.19 | 0.00 |
| Train on Physical Reasoning + Two-Cup Ablations, test on Two-Cup task | | | |
| Dataset | Correct | Incorrect | Elsewhere |
| Two-cup task | 0.48 | 0.50 | 0.02 |
| Freebie | 1.00 | 0.00 | 0.00 |
| Known Specific Location | 1.00 | 0.00 | 0.00 |
| Known Cup Location | 0.94 | 0.03 | 0.03 |
| Outside (visible) | n/a | 0.00 | 1.00 |
| Outside (occluded) | n/a | 0.12 | 0.88 |

Table 1: Results on the Two-Cup task and the control datasets. The models' predictions before the critical frame are always split roughly evenly between the two cups, when relevant. In the Outside experiments, "elsewhere" is the correct answer, because the ball is outside of both cups.

normal. This should give away the ball's location before any inference is necessary. Similarly, in the latter, to ensure that the model can track an occluded ball with only general knowledge of its location (i.e. it identifies that the ball is in a cup, but not exactly where), we reorder the frames such that the small occluders rise, then the ball falls, then the scene proceeds. Finally, to ensure that the model is tracking the ball and not just guessing a cup, we create the **Outside** control set. In this set we use the Two-Cup setup, but the ball falls entirely outside of the cup-and-wedge structure. There are two settings: **occluded**, where it falls behind a third occluder, and **visible**, where it is never obscured.

The experimental model succeeds with high accuracy (88%-100%) on each of the control sets, as shown in Table 1. More detailed explanations for the results are in the Appendix.

**Two-Cup Task Experiment**   As shown in Table 1, the predictions of the experimental model (Physical Reasoning + Two-Cup Ablations) do not change after the critical frame. In each test video, the occluder falling to reveal the empty space does not cause the experimental model to update its prediction. Comparatively, when the model is trained directly on the Two-Cup task, it predicts the correct cup after the critical frame roughly 81% of the time. These results show that the neural network model trained only on object tracking within the physical constraints of the Two-Cup setup does not learn an implicit representation of the logical operators required to pass the Two-Cup task.

### 4.2   EXPERIMENT 2: ZERO-SHOT TRANSFER FROM REASONING-BY-EXCLUSION TASK

One explanation for the failure observed in the above experiment is that the models never encountered a situation in which the ball could be in either of two, non-contiguous locations at training time, and thus never learned to represent and reason about this form of uncertainty, or to update its beliefs about the location of the ball once the uncertainty is resolved (as the ADEPT model from §3.2 was able to do). To test this possibility, we provide the model with much richer training data, training it on the logical structure of the task without giving away the visual details of the solution. We train an OPNet model not only on our original Physical Reasoning + Two-Cup Ablations data, but also on Invisible Displacement data, which requires the same inferences as the Two-Cup task but uses a visually different setup. We then observe that model's performance on the Two-Cup task.

**Control Experiments**   Similarly to the Physical Reasoning + Two-Cup Ablations experiments, we also introduce control experiments in order to rule out the possibility that the model fails to learn the necessary object tracking. We test the Physical Reasoning + Two-Cup Ablations + Invisible Displacement model on all of the earlier control sets, plus three designed to test the model's ability to learn from the Invisible Displacement training task– **Invisible Displacement Test**, **Invisible Displacement Freebie** and **Roll-then-occlude**. Invisible Displacement Test is held-out test data. In Invisible Displacement Freebie, to rule out failure in the object tracking required for the Invisible Displacement task itself, when the first occluder drops in the Invisible Displacement task, the ball is found. Finally in Roll-then-occlude, similarly to the "Known Specific/Cup Location" experiments, we test to see that the model can track an observed ball behind occlusion. The ball rolls and comes to a stop before the occluders rise, and then the scene progresses like normal.

The model succeeds at the control experiments, (including the Invisible Displacement test), scoring between 95-100% accuracy on all datasets. Detailed results can be found in the Appendix.

**Two-Cup Task Experiment**   Again, as in the prior experiment, the model performs at chance on the Two-Cup test (details in Appendix). The neural model makes the same errors as in the other tests: when the occluder falls, it does not update its prediction. Even though the model is trained on examples of the exact reasoning pattern on which it is tested, it is still unable to generalize to the Two-Cup task. There are at least two possible explanations. First, the failure might suggest that the model did not learn to solve the Invisible Displacement task in the "right way", and thus despite learning to solve that task at training time, the learning does not result in representations that are reusable on the Two-Cup task. A second explanation is that the representations learned *do* capture the logical structure of the task, but the model is not able to recognize the similarity between the tasks in the zero-shot transfer setting. This latter failure would suggest that the neural network could succeed with additional training focused on recognizing visually novel instantiations of the same logic.

### 4.3   Experiment 3: Transfer Learning Across Reasoning-by-Exclusion Tasks

In our final experiment, we test whether the neural models that failed to reason by exclusion might nevertheless be learning some representation relevant to the logical structure of both tasks. If they do, pretraining on Invisible Displacement and then finetuning on the Two-Cup task might allow it to succeed more quickly at the held-out Two-Cup test than training on the Two-Cup task alone. To examine this, we compare how quickly OPNet models begin to succeed on the Two-Cup task when first trained on Invisible Displacement or Physical Reasoning to models initialized from scratch.

Figure 4 shows the results. We find that the model that is first trained on Invisible Displacement and then trained on the Two-Cup training data succeeds at the held-out Two-Cup test set before both a model that is trained on the Two-Cup task from scratch and a model trained only on object tracking.

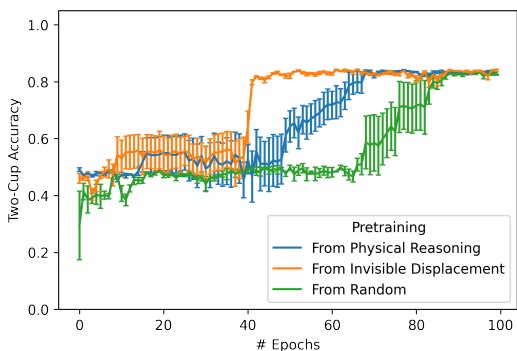

Figure 4: Accuracy (% of examples that place bounding box for the ball in the "correct cup") on the Two-Cup task over time when finetuned on the Two-Cup task and pretrained using various methods. Each is averaged across 5 random seeds. There are 5,000 data points in one epoch.

## 5   Discussion

This work asks whether neural network models are capable of learning *implicit* representations of logic from self-supervised training in the domain of physical reasoning. Overall, the results in Section 4 are primarily negative. We see in §4.1 that the neural networks trained on a basic physical reasoning task fail to generalize to the Two-Cup task. More importantly, in §4.2, we see the same models failed even when trained on the Invisible Displacement task, a task which should in principle rely on exactly the same conceptual representations as does the target Two-Cup task. Thus, the evidence strongly suggests that the neural networks are learning to solve the training tasks using representations other than those which the tasks are designed to require. However, in Section 4.3, we see suggestive evidence that the neural network might learn some desirable representations related to logical reasoning, demonstrated by the fact that training on one logical reasoning tasks speeds

learning on another logically identical task. While this falls short of our test for possession of an (implicit) logical representation, it does indicate that the network may be capable of developing the desired representation, e.g., under different training conditions.

As a proof-of-concept, the ADEPT model we analyzed in §3.2 *was* able to succeed on the Two-Cup task. This indicates that, at a minimum, a good model of the physical dynamics of the environment combined with a good model of possible worlds is sufficient to deploy implicit logical operators for reasoning. Thus, when we observe that the object tracking neural network models fail on our evaluations, it implies that they are failing to learn at least one of the relevant components of ADEPT's model of intuitive physics. Our control conditions presented in Sections 4.1 and 4.2 suggest the model learns to track objects throughout the environment reasonably well. Thus, we are inclined to conclude that the neural model fails because it does not learn to represent uncertainty over multiple possibilities and/or lacks mechanisms for updating beliefs about possibilities in response to observations. Future work considering alternative architectures and loss functions may well reveal more positive results.

## 6 RELATED WORK

Intuitive physics is thought to be one of the domains of "core knowledge" in human cognition, which are the foundations of higher-level mental processes (Spelke & Kinzler, 2007; Carey, 2009). In recent years, computational approaches to intuitive physics problems have gained popularity– Duan et al. (2022) have conducted a comprehensive survey. We focus on models with object-centric representations (Smith et al., 2019; Shamsian et al., 2020).

There are several datasets of intuitive physics tasks for computational models to train and test on. IntPhys 2019 is a benchmark dataset, on which models are trained only on positive examples; the test task is then to distinguish examples that do and do not conform to the physics of the environment (Riochet et al., 2018). Published concurrently is (Piloto et al., 2018), which takes a similar approach. Rather than distinguishing consistent/inconsistent image sequences, the PHYRE task is a dataset of physical reasoning tasks where the objective is to take one action to manipulate the scene and reach a goal state (Bakhtin et al., 2019). Within the realm of object tracking in video, the CATER dataset contains videos of a target object as it is occluded and shuffled behind and within other objects in a three-dimensional scene (Girdhar & Ramanan, 2020). Although it is not an intuitive physics task per se, similarly to this work, the ACRE dataset borrows the "blicket detector" task from developmental psychology in order to measure the ability of neural network-based and neurosymbolic reasoning models at the task of *causal induction*, or inferring causality from limited data (Zhang et al., 2021).

Neural network models are commonly thought not to be able to learn representations of explicit logical operators that generalize flexibly (Marcus, 2001). Some work pursues that question empirically, and seeks to determine whether, given a fixed set of symbols, the functions of explicit logical operators can be learned or distinguished from one another (Evans et al., 2018; Traylor et al., 2021). Other work seeks to determine whether these models can generalize negation to novel symbols (Hill et al., 2020). In the field of natural language processing, large neural language models are often trained on large corpora of text, and then tested on natural language inference tasks, or those requiring negation (sentences containing words such as *no* and *not*, which is an analogous framework to the implicit logic testing presented in this work. The representation of negation within these models tends to be dependent on the sentential context (Kassner & Schütze, 2020; Talmor et al., 2020).

## 7 CONCLUSION

In this work, we measured neural models' ability to learn implicit representations of negation and disjunction from data. We found that, despite tracking objects as they move and are occluded throughout the environment, the neural models were not able to generalize to a diagnostic task requiring implicit negation and disjunction without observing examples from that diagnostic task itself. Furthermore, when trained on visually dissimilar data that requires the same logical capabilities, transferring the representations to the target task did not improve performance. However, when training on the diagnostic task, pretraining on a similar logical task caused the models to learn the target task faster than other initialization methods. These results show a potential weakness of models that attempt to learn and generalize logical capability from data.

## 8 ACKNOWLEDGEMENTS

This research is supported in part by DARPA via the GAILA program (HR00111990064). The views and conclusions contained herein are those of the authors and should not be interpreted as necessarily representing the official policies, either expressed or implied, of DARPA or the U.S. Government. We would also like to thank James Tompkin, George Konidaris, Tomer Ullman, Kevin Smith, Chen Sun, Jack Merullo, Nihal Nayak, and the participants of the McDonnell Workshop for their helpful comments.

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
