# OpenReview forum: "Can Neural Networks Learn Implicit Logic from Physical Reasoning?"
_ICLR.cc/2023/Conference — ICLR 2023 poster_

### Official Review · Reviewer_9VQw · 2022-10-22

**Confidence:** 4
**Correctness:** 3
**Technical Novelty And Significance:** 2
**Empirical Novelty And Significance:** 3
**Recommendation:** 6

**Clarity, Quality, Novelty And Reproducibility:**

The paper proposes a novel machine learning task, the overall clarity of this paper is above average. The quality of this paper is good, and the reproducibility could be improved if the dataset and codes can be published.

**Strength And Weaknesses:**

### Strengths:
- The authors has designed many experiments and ablation datasets, which have covered different aspects in cognition for learning logic reasoning.
- The paper is generally well writen, all the details are covered either in the main text or in the appendix. The authors also provide enough discussion content to explain the observations.
- The paper targets answering an important problem in AI.

### Weaknesses:
- Although the motivation of this paper is intriguing, the methodology adopted by this paper might be weak, because it is difficult to evaluate neural networks' logical reasoning ability by just an empirical study on one type of task. Results on one or two tasks could not prove that they can represent or learn logical calculus theoretically.
- The comparing approach has a built-in Bayesian modelling module, which is too strong in my opinion. The conclusion could be more convincing if the authors can choose a learning system enhanced with explicit symbolic representation, e.g., Neuro-Symbolic Systems.
- It would be more interesting if the authors could provide a set of experiments with preverbal children or animals using the same set of training data. I think the number of training examples could be drastically reduced compared to deep learning models.
- I appreciate that this paper provides many datasets and experiments to make sure the empirical study covers every possible aspect of this problem, the naming of datasets and organisation of experiments are a bit messy. Maybe the reading experience could be improved by adding a graph of the datasets/tasks illustrating their relations, instead of hiding them in the main text.

**Summary Of The Paper:**

This paper tries to study if deep neural networks (with a built-in representation of object physics) can learn logical relations such as negation and disjunction. Inspired by cognitive science, the authors propose a task called Two-Cup which requires the agent to have the ability of reason by exclusion. This work has designed extensive experiments and ablation studies, the results show that the tested neural network failed to learn the target concepts.

**Summary Of The Review:**

This paper trys to answer an important problem in AI research, the authors have provided an interesting task and extensive emprical study to answer this question. The experimental methodology could be improved and the result is not strong enough.

---

> ### Author Response · Authors · 2022-11-18
> **Response to reviewer 9VQw**
>
> Thank you for your thoughtful review of our paper. To address your concerns;
>
> * Diversity of training experiments: Please see the meta-comment for more discussion.
> * Comparison to a Bayesian model: The use of ADEPT was not meant as a baseline or direct comparison to the neural model. Rather, it was included as a proof of concept: to demonstrate that the Two-Cup task could be solved by a model of vision that did not have symbolic logic (in the form of an explicit representation that the ball is in Cup A or Cup B, and that it is subsequently not in Cup A) integrated into its decision making process. We use this model because, a priori, it might have been the case that even a perfect physics engine would not necessarily solve our target task. If that were the case, it would not be fair to expect our neural models to do so. Thus, this was included to clarify what we meant by “success” in our context–i.e., what would a model with implicit logic look like and how might it succeed. It is not meant to be a direct competitor to the neural network models in the usual sense in which models are compared to one another in machine learning.
> * Comparisons to developmental psychology experimental results: We cite Feiman, Mody, & Carey (2022), who show that 17-month-olds spontaneously succeed in the 2-cup task with no training and review evidence that 18-month-olds spontaneously succeed on Invisible Displacement tasks without training, as well. We clarify this connection in our paper in Section 2.1.
> * Experimental Clarity: Please see the meta-comment for discussion.

---

### Official Review · Reviewer_mWJf · 2022-10-25

**Confidence:** 3
**Correctness:** 3
**Technical Novelty And Significance:** 3
**Empirical Novelty And Significance:** 3
**Recommendation:** 6

**Clarity, Quality, Novelty And Reproducibility:**

The quality and clarity of the paper are good. The descriptions about the test hypotheses and experiments are generally clear and easy to follow. The originality is also good by considering that by now (as far as I know) there is no study on the logical reasoning ability of neural models in the domain of intuitive physical understanding, although similar studies exist in other domains. There is no code or data provided in the supplemental material, thus I cannot reproduce the reported experimental results by now.

**Strength And Weaknesses:**

Strengths:

(1) A new study on the logical reasoning ability of neural models in the domain of intuitive physical understanding.

(2) Carefully designed experiments conducted to prove or refute some interesting hypotheses.

Weaknesses:
The study is restricted on a relatively small domain where the results are entangled with computer vision processing. The findings are hard to adapt to general domains about the reasoning ability for classical logics. In particular, the findings do not tell whether neural models have logical reasoning ability when computer vision processing is not needed.


**Summary Of The Paper:**

The paper studies the problem of whether a neural network model can learn an implicit representation of negation and disjunction in the domain of intuitive physical understanding that predicts how objects might move in space based on visual input. Three experiments are carefully designed to draw two conclusions: on one hand, neural models cannot generalize to a diagnostic task requiring implicit negation and disjunction without observing examples from that diagnostic task; on the other hand, when training on the diagnostic task, pretraining on a similar logical task can enable the models to learn the target task faster than other initialization methods.

**Summary Of The Review:**

The paper studies an interesting and important problem on whether a neural model can do perform implicit logical reasoning in the domain of intuitive physical understanding. Some findings are made through carefully designed experiments. However, the findings do not tell whether neural models have logical reasoning ability when computer vision processing is not needed.

---

> ### Author Response · Authors · 2022-11-18
> **Response to reviewer mWJf**
>
> Thank you for your review of our work. Please see the meta-comment in regards to the domain-specificity of our experiments.

---

### Official Review · Reviewer_1KAw · 2022-10-27

**Confidence:** 3
**Clarity, Quality, Novelty And Reproducibility:** The paper is easy to follow and the i…
**Correctness:** 4
**Technical Novelty And Significance:** 3
**Empirical Novelty And Significance:** 3
**Recommendation:** 6

**Strength And Weaknesses:**

Strengths:
A number of training modes were designed and detailed experiments were conducted to comprehensively evaluate the ability of neural network to learn implicit representations of negation and disjunction from data. It has a strong reference significance for works in logical reasoning in neural networks.

Weaknesses:
There are too few experimental charts, thus the experimental process is not clear and intuitive enough. Another obvious problem with this paper is lack of sufficient explanation of the results. You need to explain your simulation results in detail and why you got such results. In page 9, DISCUSSION, “Thus, we are inclined to conclude that the neural model fails because it does not learn to represent uncertainty over multiple possibilities and/or lacks mechanisms for updating beliefs about possibilities in response to observations. ”, this explanation is ambiguous and not convincing.


**Summary Of The Paper:**

Summary: The paper mainly explores whether the neural network can learn the implicit representations of negation and disjunction. The results of experiment shows that the neural network lack the ability to generalize to task that requires implicit logic.
Contributions:
1)The authors introduce the notion of implicit logic into the repertoire of neural network evaluation.
2)The authors use a test of logical inference in humans to evaluate the neural network.
3)The experiments offer some suggestive evidence about the models’ ability to transfer representations between logically equivalent tasks.


**Summary Of The Review:**

The research work of this paper mainly focus on neural network’s ability of learning implicit representations. Overall this paper is a comprehensive and complete work.
If the above problems are well-addressed, this reviewer believes that the essential contribution of this paper are important for research of neural models’ ability to learn implicit representations.

---

> ### Author Response · Authors · 2022-11-08
> **Clarification Questions about Review**
>
> Thank you for your comments. We'd love to address them but are not sure how to do so given the details you provide. Could you please clarify the following?
>
> 1. You say "the experimental process is not clear and intuitive enough" and request more charts. Could you be more specific about which sections are unclear and would benefit from more charts?
>
> 2. You say there is a "lack of sufficient explanation of the results" and cite one example. Is this the only example you found to be insufficiently explained? If so, we can definitely provide more details justifying this conclusion. But if you believe the issue is more pervasive, we'd appreciate more explanation on why you think that is the case so we can address the concern.

---

> ### Author Response · Authors · 2022-11-18
> **Response to reviewer 1KAw**
>
> Thank you for your comments and for your review of our paper. In regards to your comment about clarity, please see the meta-comment, as well as the Appendix where we have added additional clarifying figures.

---

### Official Review · Reviewer_7YcZ · 2022-10-29

**Confidence:** 4
**Correctness:** 4
**Technical Novelty And Significance:** 3
**Empirical Novelty And Significance:** 3
**Recommendation:** 8

**Clarity, Quality, Novelty And Reproducibility:**

The paper is very well written and easy to follow. The only suggestion I would do in this regard is to include further details and a figure about OPNet in the appendix for the paper to be self-contained.

The paper is sound and detailed in their experiments. The methodology to measure implicit negation and disjunction is also novel, which would help to increase the understanding of the research community on the ability of NNs to generalise implicit logic operators if it weren't because it only seems to confirm the previous works.

And I want to remark the "seems" because that is preventing this work from really doing a remarkable contribution. The results suggest that the missing thing could be a higher variety of implicit logic examples as in those previous works but as it is right now we can't know for certain. I know that this additional experiment is not trivial, and probably -and sadly- the rebuttal period won't be enough to carry it out. But without that, the conclusion of this work are only suggestions when built on top of that previous literature.

Once that experiment is included, and we are able to see if that is really a key ingredient in this setting as well, I believe this will be a significant contribution and I strongly encourage the authors to continue this line of work.

**Strength And Weaknesses:**

The paper follows a recurrent and very important topic on the ability of NNs to do compositional generalisation of logic reasoning in a novel set up. One of the main strengths of the paper is the careful detail of the experiments and control tests -which rule out plausible concerns about if the model is really failing to solve the task due to the inability of generalisation of logic reasoning or something else-.

My main concern about the paper is that, while authors cover a good amount of related literature, they missed several key papers about compositional generalisation in reinforcement learning. I strongly encourage authors to check those earlier studies because they would explain why the NNs fails to generalise in this work, and what should have been done (or at least tried) for the model to past this test:

* Hill et all, 2020 did an experiment on the ability of DRL agents to generalise negation to unseen instructions. In that work the agent is trained with positive instructions such as "find a plane" or "find a ball" and the agent should find those object in a room and stare at them for a few second to succeed. For some of those objects the agent was trained with negation with instructions such as "find something that is not a plane", and then evaluated to do  find something that is not X, where x was a training object that had been used for positive instructions only. Hill et al. found that for the agents to succeed, they required to had been trained with at least 100 negated instructions to pass this test.

* Also in that line, León et all. did a similar experiment in 2D scenarios to test the ability of DRL agent to achieve compositional generalisation with negation and disjunction and found that with object-based encoders - that this paper under review is using already-  and early stopping, 6 objects are enough to generalise negation and disjunction.

I believe that those two works study a form of implicit logic in a similar fashion that this paper does, but even in the latest they required 6 examples of that form of reasoning before achieving successful results. Thus, when this work shows in Section 4.1 that the agent is not doing that form of generalisation after just one example, it actually aligns with those existing works. Also the result from section 4.2 showing that the agent that has learnt to solve this kind of generalisation once is able to learn to solve the experiment faster also reinforces the hypothesis from those previous works that I cited.

Thus, I believe that it would be key for this paper to be complete to do one further test where they train the NNs with 6 different variants of implicit reasoning and then evaluate the NNs with the two cups test again. Since the setup of this work and context is so different (this is not an RL work for instance) it would be a great contribution to see if the hypothesis from those previous works are confirmed here as well.

I believe then that authors should include those two previous works in their RW section and carry an additional experiment to see if that helps the NNs to pass the test.

Hill, Felix, et al. "Environmental drivers of systematicity and generalization in a situated agent." International Conference on Learning Representations. 2020.

León, Borja G., Murray Shanahan, and Francesco Belardinelli. "Agent, do you see it now? systematic generalisation in deep reinforcement learning." ICLR Workshop on Agent Learning in Open-Endedness. 2022.

**Summary Of The Paper:**

This paper presents an in-depth study on the ability of neural networks (NN) to learn implicit logic of negation and disjunction. Specifically implicit negation and disjunction here refers to the ability to tell that if a ball fall in one of two recipients, the NN doesn't know which one, but then one is shown empty (the ball is not there) then the agent should know that the ball is in the other recipient.

Authors do not train the agent directly on the task above, which is used as test task but train in related tasks with the same objects and background. Results in control experiments show that the NN learns to track the ball but it is not able to solve the task.

**Summary Of The Review:**

Careful and detail experiment in a novel setting for a key ingredient of intelligence. However, authors missed key pieces of literature exploring the very same questions that are the main focus of this work. This leaves this paper incomplete since authors haven't checked if a key ingredient in those earlier studies is what they were missing here.


----------------UPDATE---------------------

After discussing with the authors, I agree with them that from the point of view of demonstrating that physical reasoning has arisen, but it does not mean that (implicit) logical reasoning has arisen too, this work is novel.

I still believe that the last experiments suggest that if the agent had been trained in at least 6 different contexts implicit logic reasoning might have arisen. Since Hill et al., demonstrated that agents struggled to learn logic without enough examples  and León et al.  pointed that with object-based architectures -as the one in this paper- 6 examples is enough for compositional logic reasoning to arise. I think this should be pointed in the future directions as a possible solution to tackle why the agents are failing in the experiments of this paper.

I believe that the points above need still to be better clarified/incorporated in the discussion conclusion section, but given author's response so far, I am confident that they will and updating my score accordingly.

---

> ### Author Response · Authors · 2022-11-18
> **Response to reviewer 7YcZ**
>
> Thank you for your thoughtful review of our paper, and for highlighting related work. We have updated our draft to include Hill et al. in our related work section.
>
> You ask for us to consider replicating some of the findings of this paper in our experiments. This suggestion is interesting, but upon considering it, we realized that it is not feasible because the basic goals of the papers (i.e., the types of generalization being tested) are not aligned. Thus, the type of training variation that Hill et al provides the model is not possible for our setting. We elaborate below.
>
> In Hill et al. (2020) and León et al. (2022), the model is provided with 20 or 6 examples respectively of a negation task (“Find me an object that is not the box”) and then must generalize them to a test set containing novel objects– a test of compositional generalization. That is, the goal is to train on e.g., twenty objects (e.g., {not(box), not(ball), not(dog), not(block), not(apple), not(bottle), …}) and then test on a novel object (e.g., not(pencil)). In our work, we do not test whether the implicit form of negation generalizes to novel objects, but rather test whether negation emerges in a zero-shot setting as a result of representing the physics of the environment. We don’t vary the object at all (i.e., in our experiments, it's always a ball that is falling) because this object-level generalization is orthogonal to what we are testing. Therefore, for our study, the best analogy to the “rule of 20” from Hill et al. would be to come up with 20 visually unique physical scenarios in which finding the hidden object requires using the disjunctive syllogism. Our paper already uses two such scenarios–the two cup task and the invisible displacement paradigm. As far as we are aware, inventing other such scenarios is not immediately possible without significantly altering the rules of physics (e.g., we could allow objects to teleport to enable ambiguity to arise in other scenarios, but this would have irrevocable consequences for our other experiments). Thus, instantiating Hill et al’s “rule of 20” is, we believe, not possible in our setting. Thus, while generalizing negation to novel objects is a significant achievement and is certainly related to this line of work, it is addressing a separate and complimentary question to our task.

---

> > ### Comment · Reviewer_7YcZ · 2022-11-20
> > **Response to authors**
> >
> > Thank you for your detailed response and review of related work.
> >
> > First, please note that there is not "rule of 20" in Hill et all. but "rule of 100" instead. This is why I mentioned León et al. (2022), since they found evidences with only 6 -and not only studied negation but also disjunction, which you study here-. Thus, it would be a "rule of 6", i.e., it would require you to do 5 additional visually unique physical scenarios, one you already did in Section 4.2. Way less than the 20 that you mentioned in your response.
> >
> > I completely agree with the authors that this is a different experiment setting for generalisation of negation and the other operators that were already studied in that literature. For that reason I already stated that the work has novelty in my original review.
> >
> > However, I believe that without such experiment this work is still incomplete and therefore I am keeping my original borderline score. I agree that in the setting that this experiment is conducted it is non-trivial to provide a variety of tasks. However, without that, I see this work  only as another addition to the myriad experiments (specially if we look at literature on locomotion and RL) evidencing that from on/two single tasks deep leaning agent overfits and does not learn physics, e.g., Garnelo & Shanahan (2019). Consequently, I still don't think this work meets the bar for acceptance at ICLR.
> >
> > Garnelo, Marta, and Murray Shanahan. "Reconciling deep learning with symbolic artificial intelligence: representing objects and relations." Current Opinion in Behavioral Sciences 29 (2019): 17-23.

---

> > > ### Author Response · Authors · 2022-11-22
> > > **Response to reviewer 7YcZ**
> > >
> > > Thank you again for the discussion.
> > >
> > > While we are not testing compositional generalization between different prejacents of negation (e.g. from not(ball) to not(pencil)), we agree that generalization to different instantiations of the disjunctive syllogism in the physical domain could be informative. However, it is simply not possible to run such an experiment. As we specified, there are only two known instantiations -- the 2-cup task and invisible displacement -- and we currently include both. Thus, by this criteria, no work on the disjunctive syllogism could ever be “complete”.
> > >
> > > We want to re-emphasize the novelty of the contribution in this paper, which does not depend on the experiment you describe. This paper is about whether there is a plausible mechanism by which (implicit) logical reasoning can arise by generalizing from physical reasoning. We do not make claims about neural networks failing to learn physical reasoning (as do the other papers to which you refer). In fact, within our simplified environment, the neural networks perform well on all the physical reasoning tests. We also do not make any claims about the impossibility of neural networks learning logical reasoning more generally. Rather, our claim is that, even having learned the physical reasoning tasks well -- tasks that plausibly require representing physical constraints that are sufficient to solve the reasoning task, as the experiment in section 4.1 shows -- the models fail to generalize the rules of the environment in a way consistent with implicit logical reasoning.

---

> > > > ### Comment · Reviewer_7YcZ · 2022-11-23
> > > > **Follow up**
> > > >
> > > > Thank you for the follow up. I actually agree pretty much with what you said in your latest response.
> > > >
> > > > I we look at this work from the point of view of physical reasoning has arised, does it mean (implicit) logical reasoning? I agree that I don't know of previous works making this question in the miriad I mention. Looking from this perspective, I agree that the paper has novelty despite not checking the '6 rule' derived from Hill and Leon's works. Thus, I am updating my score since my biggest concern was that.
> > > >
> > > > Still, I would believe it would be benefitial to highlight this point more clearly - about what the paper claims and what it does not claim and its novelty- in the discussion/ conclusions section. As it is written right now it is confusing. I also believe that it would be good to indicate that 'rule of 6' experiment as a future line of work.
> > > >
> > > > Nevertheless, I am updating my score taking in consideration that the authors will incorporate the amendments for the final version (I am not sure of how long they have to update the paper for the rebuttal).

---

### Official Review · Reviewer_ggkT · 2022-11-01

**Confidence:** 3
**Correctness:** 3
**Technical Novelty And Significance:** 3
**Empirical Novelty And Significance:** 3
**Recommendation:** 8

**Clarity, Quality, Novelty And Reproducibility:**

The paper is well written and is of good quality. It adds to the list of tasks for evaluating the ability of neural networks on reasoning tasks. It may be difficult to reproduce the experiments if the code is not released as designing the experiments is heavily involved. Some example animations showing the data generation process would certainly help.

There are some minor typos:

"taskk" in 3.1.2 and "to to" in the last paragraph of Section 6.
The caption of Figure 3 refers to "blue ball" but I do not see any blue ball in the figure.

**Strength And Weaknesses:**

Strength

Analyzing the limitations of neural networks is an important contribution to the field, which enables researchers to further enhance the capabilities of neural networks.
The paper proposes the first set of experiments to characterize the failures of neural networks in tasks that require reasoning. It's well known in the literature that neural network in general lack reasoning capabilities but it's rarely explicitly demonstrated in a set of experiments. This work pushes that boundary.
The proposed work is grounded in theory that aims to understand how logical reasoning "emerges" from human and non-human animals alike.

Weaknesses

Fine-tuning experiments, although in a logical reasoning context, are not really revealing anything new that we don't already know about neural networks.
The experiments are only limited to one type of logical reasoning task and neural network architecture.
It's not clear what it means for a neural network to succeed in solving the Two-Cup task. The ADEPT baseline seems to predict the correct cup 100% of the time but the neural network is considered successful by predicting the correct cup 81% of the time.

**Summary Of The Paper:**

The paper proposes an experiment for evaluating whether neural models can learn to represent logical reasoning operators purely from observations. The experiments are taken out of literature in developmental and comparative psychology, in which a neural network is trained to represent negation and disjunction operators. The focus is on learning implicit representation, which are domain-specific and may not necessarily generalize to other domains but nonetheless measure the ability of neural models to implicitly perform logical reasoning in some limited fashion.

The experiments reveal that neural networks are indeed not able to learn to represent logical operators without being trained directly on test data. However, the transfer experiments show that neural networks do learn some useful representations from one domain with similar logical task but different features, that enable them to quickly learn to perform well on a test task.

**Summary Of The Review:**

More research work highlighting the failures of neural networks and drawing insights on how to improve them, is needed. This work adds to the list of benchmarks of highlighting the limitations of neural networks on logical reasoning tasks. While more experimentation on other types of reasoning tasks and for a wide range of neural architectures is still needed, this work will open up more opportunities for other works to follow in the same vein.

---

> ### Author Response · Authors · 2022-11-18
> **Response to reviewer ggkT**
>
> Thank you for your comments. We have fixed the typos you noted. Please see the meta-review for comments about the domain specificity of our results.
>
> * Fine-tuning experiments: Our goal by running these experiments was to observe whether representations would transfer between visually dissimilar tasks that require the same logical reasoning. We were not seeking novelty of experimental design here. Rather we were interested in whether there was any evidence that  models can transfer information that is useful for logical reasoning tasks, especially within a visual domain. This transfer paradigm is one established way that our field measures representational similarity and transfer, and so was an appropriate way of measuring generalization (with perhaps less strict success criteria than our earlier experiments).
> * 81% accuracy on Two-Cup Task: We agree with you that this less-than-100% accuracy is something that warrants further investigation. In short, less than 100% accuracy seems to provide counter evidence to claims that the model is truly learning the task. This is one reason why we see the results in Section 4.3 as only weak evidence supporting the paper’s main hypothesis. That is, we only interpret this as evidence that some aspect of the representation is shared, not evidence of anything stronger than that. That being said, we did investigate further into why we see a plateau at 80%. In short, we did not find a definitive answer, but we summarize our investigation in Section F of the Appendix (“Training on Two-Cup Task”).

---

### Author Response · Authors · 2022-11-18
**Meta-comment**

We’d like to thank all of the reviewers for their time and for their thoughtful comments. Some broader points that we’d like to make:

* Domain limitations: Reviewers ggkT, mWJf, and 9VQw noted that our results about logical reasoning were limited to one domain– visual reasoning– and were concerned about whether our results would generalize to other domains. Our domain restriction is actually a deliberate choice, and part of the contribution of the paper. We look for evidence of domain-specific (i.e., implicit) logical reasoning using tests adapted from studies on humans and animals intended to test the same. We make no claims as to whether or not neural networks are able to learn and generalize abstract logic broadly, using explicit operators such as OR, AND, and NOT– this literature is contentious (See Marcus 2001) and would require experiments of an entirely different nature than what we propose here. Rather, we use these experiments in order to examine whether behavior that would be a signature of logical reasoning in a specific domain can emerge as a result of what a model has learned about that domain (in our case, visual reasoning). These experiments are thus domain-specific by design. Although our experimental framework could be extended to other domains (e.g. robotics and language processing), doing so would address a different set of questions. Following work in cognitive science on the emergence of logical operators in thought (e.g. Feiman, Mody, & Carey, 2022), we think it may be promising both methodologically and theoretically (in terms of recapitulating evolution and human development) to start with a domain-specific reasoning success and then extend that to other domains. Had we found a domain-specific success, domain-general extensions would have been natural. However, our largely negative evidence about domain-specific reasoning suggests a need to understand the source of the domain-specific limitation before extending outwards. We are of course ultimately interested in domain-general reasoning, but view this as a long-term research program that will require many (likely interdisciplinary) studies before we can make claims about abstraction or domain generality or logic.
* Additional explanation of results: Reviewers ggkT, 1KAw, and 9VQw expressed interest in clearer visual explanations of our data and results. In order to make our experimental design as clear as possible, we’ve updated our Appendix (Section A) with additional figures and tables that are designed to improve clarity.
* Code release: We intend to release all code upon publication.

Roman Feiman, Shilpa Mody, and Susan Carey. The development of reasoning by exclusion in infancy. Cognitive Psychology, 135:101473, 2022.

Marcus, Gary F. The algebraic mind: Integrating connectionism and cognitive science. MIT press, 2001.

---

### Decision · Program_Chairs · 2023-01-20

**Decision:**

Accept: poster

**Justification For Why Not Higher Score:**

The study is limited in scope and focuses on a single modality/task as highlighted by reviewers.

**Justification For Why Not Lower Score:**

N/A

**Metareview: Summary, Strengths And Weaknesses:**

The authors investigate whether neural networks can learn to implicitly perform logical reasoning out of purely perceptual information. For example, by training a network to represent negation and disjunction operators out of image data alone. The contribution is essentially of empirical nature and grounded in the literature of developmental and comparative psychology.  Experiments provide evidence that learning accurate logical operator from data alone is hard, while the network might learn some internal representation that can help transfer from different tasks of the same kind.

The reviewers agreed that addressing what kind of reasoning neural nets can learn is interesting and important. Some concerns raised during the reviewing phase include: limitation of the research question to a single task and set of operators; missing literature comparison and missing experimental details. Authors were responsive and managed to address all the major concerns, improving the quality of the paper and raising one reviewer's score towards full acceptance. The only issue remaining is the scope of the analysis being limited to a single kind of tasks, which limits the generalization of the hardness claims.

I believe that this still provides enough evidence to falsify the generally spread claim that neural nets can learn everything, for all possible tasks, from data alone. Therefore I recommend the paper to be accepted.


**Note From Pc:**

if the above contains the word "oral" or "spotlight" please see: "oral" presentation means -> notable-top-5% and "spotlight" means -> notable-top-25%. As stated in our emails, we are disassociating presentation type from AC recommendations

**Summary Of Ac-Reviewer Meeting:**

N/A